# Extracting Urban Water Bodies from Landsat Imagery Based on mNDWI and HSV Transformation

Liwei Chang [1], Lei Cheng [1,*], Chang Huang [2,3], Shujing Qin [1], Chenhao Fu [1] and Shiqiong Li [1]

1 State Key Laboratory of Water Resources and Hydropower Engineering Science, Wuhan University, Wuhan 430072, China
2 College of Urban and Environmental Sciences, Northwest University, Xi'an 710127, China
3 Shaanxi Key Laboratory of Earth Surface System and Environmental Carrying Capacity, Northwest University, Xi'an 710127, China
* Correspondence: lei.cheng@whu.edu.cn

**Abstract:** Urban water bodies are critical for sustainable urban ecological and social development. However, the complex compositions of urban land cover and small water bodies pose considerable challenges to urban water surface delineation. Here, we propose a novel urban water extraction algorithm (UWEA) that is efficient in distinguishing water and other low-reflective objects by combining the modified normalized difference water index (mNDWI) and HSV transformation. The spectral properties of urban land covers were analyzed and the separability of objects in different color spaces was compared before applying the HSV transformation. The accuracy and robustness of the UWEA were validated in six highly urbanized subregions of Beijing, Tokyo, and New York, and compared with the mNDWI and HIS methods. The results show that the UWEA had the fewest total errors (sum of omission and commission errors) for all the validation sites, which was approximately 3% fewer errors than those of the mNDWI and 17% fewer errors than those of the HIS method. The UWEA performed best because it was good at identifying small water bodies and suppressing reflective surfaces. The UWEA is effective in urban water monitoring and its thresholds are also robust in various situations. The resulting highly accurate water map could support water-related analyses. This method is also useful for scientists, managers, and planners in water resource management, urban hydrological applications, and sustainable urban development.

**Keywords:** urban water extraction; Landsat; water index; HSV color space; threshold stability; M-statistic test

## 1. Introduction

Surface water is one of the principal land cover types that supports urban sustainable development activities [1], particularly in urban water resource management, urban landscape and ecosystem maintenance, and mitigating urban heat-island effects [2–4]. Satellite imaging offers a unique opportunity to quickly monitor the variations in urban surface water, and relative analysis is becoming increasingly convenient as satellite data become more readily available and cloud computing platforms advance [5]. The accurate mapping of urban surface water is critical for urban planning [6], flood management [7,8], and water quality analyses [9,10]. Because many studies have attempted to eliminate water surfaces before obtaining pure impervious surface maps, the accuracy of urban water maps may impair the detection of other urban components such as impervious surfaces [11,12]. Several satellite products have been used for urban water extraction. In particular, Landsat data have been widely used because of an open data policy, medium-to-high resolution (~30 m), and long time-series archives [13,14].

Various water mapping approaches have been proposed over the last few decades, with water indices being the most commonly used, including the normalized difference water index (NDWI) [15], the modified normalized difference water index (mNDWI) [16],

and the automated water extraction index (AWEI) [17]. Thresholding is one of the most critical challenges faced when using water indices to extract water bodies [1]. To date, these water indices have been employed with varying degrees of success in urban settings. Yang and Chen [18] assessed different water indices to extract urban water from Sentinel-2A imagery and discovered that the mNDWI and AWEI performed better and had comparable accuracy. However, extracting water from urban areas presents major obstacles due to the complex land surface composition and sporadic distribution of small water bodies. No known water index can achieve total separability between water and noise in the obtained water maps [19]. This is due to two main reasons: (1) the spectral reflectance of narrow rivers or small ponds is easily confused with the surrounding objects or aquatic plants [20,21], and (2) some ground materials and structures, such as extremely high-albedo surfaces (reflective material roofs) and low-albedo impervious surfaces (pitch and shadows), exhibit spectral characteristics comparable to those of water [12,22–24]. Although automatic thresholding schemes have been applied, the development of robust thresholds for new water indices is still necessary.

Researchers have proposed methods for improving the accuracy of water extraction in urban regions. Yang et al. [19] proposed a noise-prediction technique to eliminate misclassified non-water areas from water maps. Yao et al. [21] generated a dark-building-shadow prediction model using the support vector machine method with the green band and NDWI as inputs. However, these classification models are complex and time-consuming [25]. Water detection and dark surface separation have both been found to benefit from color space transformation [26–29]. The traditional RGB color space, a color additive, and a mixing model employ the superimposition of three primary colors to create other colors. Invariant color spaces, such as HSV and HIS, decouple the hue, color purity (saturation), and brightness (value), allowing land cover types to be distinguished based on color differences [30–32]. Jiang et al. [33] extracted water from the background of vegetation and built-up areas using the normalized difference built-up index (NDBI), normalized difference vegetation index (NDVI), and mNDWI in the HIS transformation. Pekel et al. [31] employed the HSV transformation on Moderate-Resolution Imaging Spectroradiometer (MODIS) to reveal water area variations in a time series. Furthermore, Ma et al. [34] designed a normalized saturation value difference index in the HSV color space to detect shadows in urban areas. Ngoc et al. [35] developed an algorithm to extract water from Landsat-8 Operational Land Imager (OLI) and Sentinel-2 Multispectral Instrument (MSI) images using saturation to discriminate between the water and shadow pixels. Existing water indices are concise and efficient in water extraction on a large scale but less acute in a densely built-up city because of the water body fragmentations and complex surface compositions. Although different bands were chosen to conduct the HSV transformation, the HSV color space was certified to be suitable for the analysis and segmentation of complex images. In a spatially heterogeneous urban area, considering a combination of multiband information and the HSV transformation is a useful complement to conventional water extraction methods. As previous studies have not focused on using HSV transformations in urban water delineation based on Landsat imagery, we attempted to explore this aspect.

Therefore, this study attempts to investigate the use of HSV color space in conjunction with traditional water indices for open water extraction in urban areas. A novel urban water extraction algorithm (UWEA) is introduced and the spectral characteristics of different urban land components are studied using the established urban spectral library. The proposed algorithm aims to enhance the accuracy of urban water extraction while being robust under various environmental conditions. A comparative study was conducted in three highly urbanized cities between the UWEA and two adapted methods, the mNDWI and the Jiang et al. [33] method; the UWEA demonstrated considerably improved accuracy. Further analysis indicated that the UWEA has a more robust performance within the neighborhood of the default thresholds and achieves better separability of water and non-water objects than comparative methods.

## 2. Study Area and Data

The proposed UWEA method was developed and validated in a series of typical cities (Figure 1). Five cities with varying environmental conditions were selected for sampling: Beijing, Shanghai, Suzhou, Hangzhou, and Tokyo. Beijing is located inland and has a warm temperate monsoon climate. Shanghai is a coastal city with a subtropical monsoon climate. Suzhou and Hangzhou have the same climatic region as Shanghai but are located further inland. Tokyo is a coastal city with a subtropical monsoon climate. All five cities have dense urban areas and variable water bodies. Because of the complex compositions of these sites, extracting water from satellite images quickly and accurately is a challenging task. Scenes from Beijing and Tokyo taken at different periods (Table 1), as well as scenes from New York, were utilized to validate the accuracy and robustness of the UWEA in variable environments. All three validation scenes contained many impervious surfaces and different water bodies. Beijing features artificial lakes and wetlands, Tokyo has compact buildings and extremely dark shadows cast by high-rises, and New York features high-albedo objects.

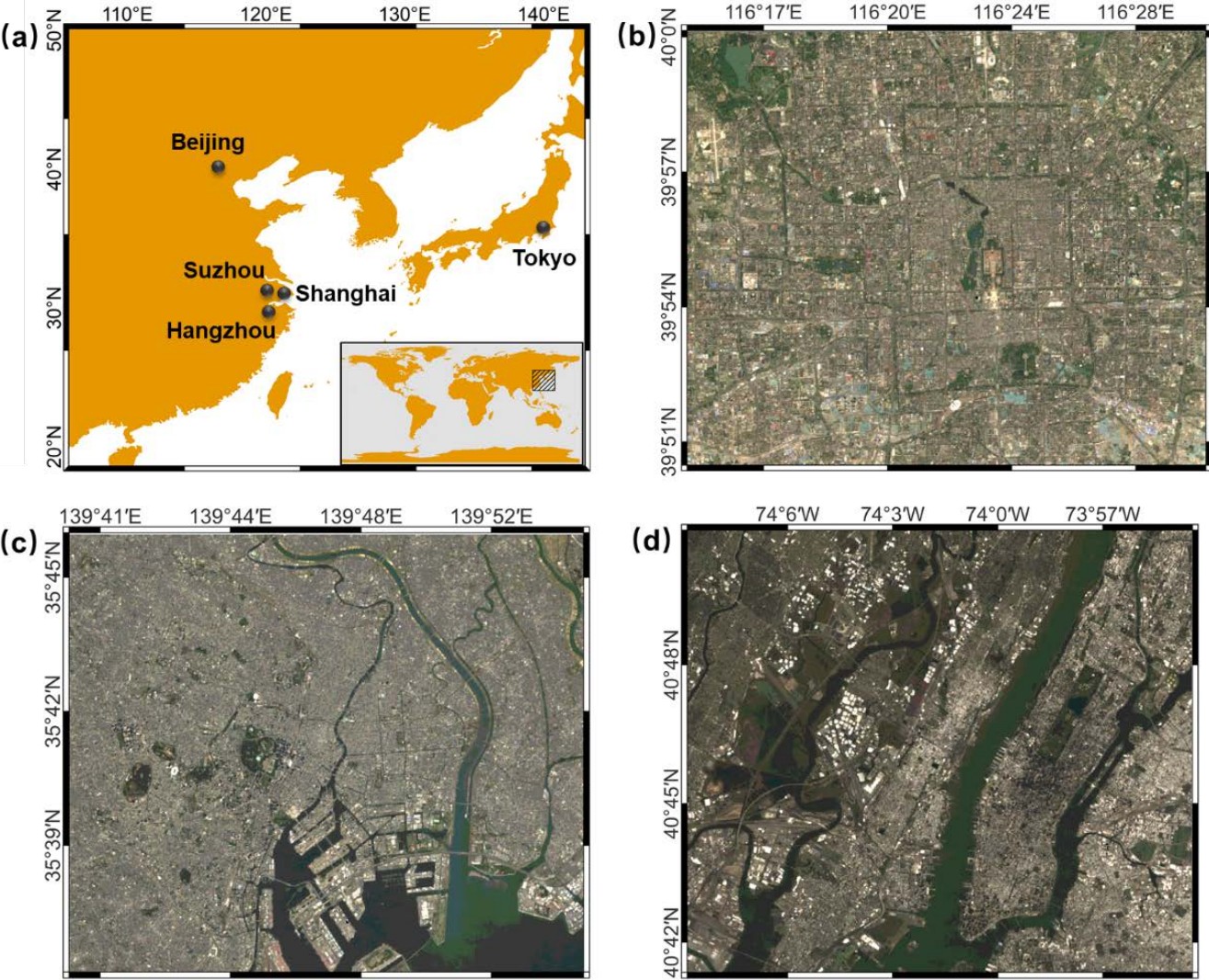

**Figure 1.** Subplot (**a**) is the location of the five sampling cities: Beijing, Shanghai, Hangzhou, Suzhou, and Tokyo. Subplots (**b**–**d**) are true color images of three validation scenes in Beijing, Tokyo, and New York, respectively.

**Table 1.** Acquisition times of images for sampling and validation.

| Dataset | City | High-Resolution Image on Google Earth (Image ©2022 Maxar Technologies) | Landsat Image (Landsat 5 TM, Landsat 7 ETM+ and Landsat 8 OLI) |
|---|---|---|---|
| Sampling | Beijing | 1 October 2019 | 1 January 2019–1 January 2020 |
| | Shanghai | 10 December 2019 | 1 June 2019–1 June 2020 |
| | Hangzhou | 28 October 2018 | 1 January 2018–1 January 2019 |
| | Suzhou | 15 March 2009 | 1 January 2009–1 January 2010 |
| | Tokyo | 30 November 2018, 24 May 2019 | 1 June 2018–1 June 2019 |
| Validation | Beijing | 28 August 2020 | 1 January 2020–1 January 2021 |
| | Tokyo | 1 November 2019 | 1 January 2019–1 January 2020 |
| | New York | 24 May 2020 | 1 January 2020–1 January 2021 |

Open and unrestricted access Landsat data play an important role in water body delineation because of their 30 m spatial resolution and exceptionally long service time. However, the available images of a certain date are very limited owing to cloud interference, and the reflectance of objects varies with time. For each city, all available surface reflectance images from Landsat-5 Thematic Mapper (TM), Landsat-7 Enhanced TM Plus (ETM+), and Landsat-8 OLI were used. The urban compositions remained stable for a short period of time and there was no considerable urban flooding throughout the study phase. Therefore, the composed map is defined as an annual average image incorporating all of the images from that year, which were selected by a time filter from the Google Earth Engine. The annual change in the surface water area is a key parameter in urban management [36,37]. Available high-resolution Google Earth images from the adjacent dates were used as real land cover data. The consistency between the two paired images was assessed via visual inspection. Table 1 shows the acquisition time information of the Landsat images and the corresponding high-resolution images.

An urban sample dataset was established for spectral analysis. Urban land components were classified into six types: water, buildings, vegetation, bright roofs, roads, and shadows. Vegetation samples were obtained from city parks and wetlands. Architectural structures and buildings under construction were defined as the building sample points. Bright roofs were high-albedo objects, particularly those made of glass and white metals, among other materials. The established sample dataset comprised 4170 points including water (1239), buildings (1518), vegetation (605), bright roofs (219), roads (351), and shadows (238). The sample points were randomly selected by visually interpreting high-resolution Google Earth images. To decompose the mixed spectral signatures, all city rivers with a width of more than 30 m (due to the spatial resolution of the Landsat image of 30 m) were sampled, with sample points selected in the middle of rivers to avoid mixed-edge pixels. Non-water points were sampled in large homogeneous regions of each type to avoid spectral mixture. For example, building sample points were usually selected in dense urban areas and road sample points were mainly selected on major roads. To deal with the variations in the shadow areas, the shadow sample points were collected within a mask of SWIR < 0.1. The number of sample points was determined by the area proportion of each land cover type on a city map.

Landsat images were used to determine the spectral reflectance of the sample points. Figure 2 depicts the spectral reflectance curves for the six land cover types. The curves represent the median reflectance values for each type. Notably, the reflectance of bright roofs was the highest among the six types throughout all six bands and it was greater than that of water. The reflectance of vegetation was particularly high at the near-infrared (NIR) band, making it easily distinguishable from others using NIR-based indices (e.g., NDVI). Buildings, roads, and shadows had similar spectral forms to water, with the exception that their values in the infrared bands, NIR, short-wave infrared 1 (SWIR1) band, and short-wave infrared 2 (SWIR2) band were greater. The most noticeable feature of water was that the values of each band were almost the lowest, particularly its strong absorption

in infrared wavelengths, making it distinguishable. Nevertheless, the reflectance value in the NIR band for water was particularly high in these cities, which was different from the spectral signatures of natural water bodies [1]. A higher NIR brings water closer to the impervious surface in the spectral form, making it more difficult to discern from vegetation and impervious surfaces. A possible explanation is that the water in urban areas contains more aquatic plants and algae due to eutrophic sewage released from urban production and households compared to natural sources of water [38,39]. The reflectance value of shadows was close to that of water in all bands, especially in the blue, green, and red bands, which may lead to the poor separation of water bodies based on these three bands only. The differences in the infrared bands may help to separate water and shadows. Moreover, the mNDWI was also effective at separating them using the normalized difference of the green and SWIR1 bands because the reflectance of water in green is always higher than that in SWIR1, and is lower for shadows.

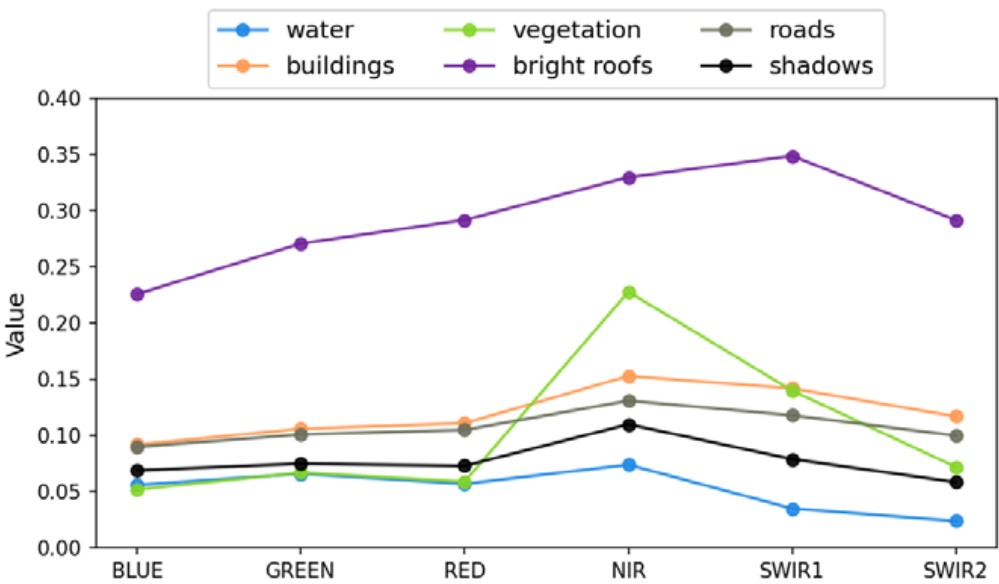

**Figure 2.** Spectral reflectance curves of six land cover types.

### 3. Methodology

#### 3.1. HSV Transformation

The HSV color space includes three components: hue (H) and saturation (S), indicating chromaticity, and value (V), indicating luminance. In the conventional HSV color model, H is expressed as an angle ranging from 0° to 360°, displaying color information ranging from red (0°) to green (120°) and blue (240°). S and V were values between 0 and 1, denoting the purity and brightness of the color, respectively. The conversion formulas from the RGB color space to the HSV color space are defined as follows:

$$V = \max(R, G, B) \tag{1}$$

$$S = \frac{V - \min(R, G, B)}{V} \tag{2}$$

$$H = \begin{cases} 0, & \text{if } V = \min(R, G, B) \\ \left(60° \times \frac{G - B}{V - \min(R,G,B)} + 360°\right) \bmod 360°, & \text{if } V = R \\ 60° \times \frac{B - R}{V - \min(R,G,B)} + 120°, & \text{if } V = G \\ 60° \times \frac{R - G}{V - \min(R,G,B)} + 240°, & \text{if } V = B \end{cases} \tag{3}$$

V indicates the maximum value of the three bands and can be used to distinguish between bright or dark objects; S represents the difference between the maximum and

minimum values, which is large if objects strongly reflect or absorb one kind of light; H is affected by the maximum value of the three bands. Generally, greater differences between the values of R, G, and B can bring more information to the HSV transformation. For example, objects with maximal reflectance in different bands will lead to different H, and the deviation between the maximum and minimum values will produce distinct S. Therefore, the performance of the HSV transformation is considerably impacted by the input bands. In this study, the values of the SWIR1, NIR, and red bands were employed as R, G, and B, respectively, to create color composite images and transform them into the HSV color space.

### 3.2. M-Statistic Test

The M-statistic, a statistical metric for measuring the distance between two classes using the difference of the means ($\mu$) normalized by the sum of the standard deviations ($\sigma$), may be used to quantify class separability [40]. The M-statistic is expressed as

$$M = (\mu_1 - \mu_2)/(\sigma_1 + \sigma_2) \tag{4}$$

The distance of mean values represents the separability of the main information of two classes, whereas the sum of the standard deviations indicates the noise contained in it. In general, a large M value indicates strong separability, suggesting a large relative distance between the two classes. The above formula was applied to scalars. For hyperspaces, M may be computed using Euclidean distance and the mean and standard deviations are vectors. The RGB color space variables and operations are as follows:

$$\vec{\mu}_1 = \left(\mu_1^r, \mu_1^g, \mu_1^b\right), \ \vec{\sigma}_1 = \left(\sigma_1^r, \sigma_1^g, \sigma_1^b\right), \ \vec{\mu}_2 = \left(\mu_2^r, \mu_2^g, \mu_2^b\right), \ \vec{\sigma}_2 = \left(\sigma_2^r, \sigma_2^g, \sigma_2^b\right) \tag{5}$$

$$M = \frac{\left|\vec{\mu}_1 - \vec{\mu}_2\right|}{\left|\vec{\sigma}_1 + \vec{\sigma}_2\right|} = \frac{\sqrt{\left(\mu_1^r - \mu_2^r\right)^2 + \left(\mu_1^g - \mu_2^g\right)^2 + \left(\mu_1^b - \mu_2^b\right)^2}}{\sqrt{\left(\sigma_1^r + \sigma_2^r\right)^2 + \left(\sigma_1^g + \sigma_2^g\right)^2 + \left(\sigma_1^b + \sigma_2^b\right)^2}} \tag{6}$$

The M-statistic was used to quantify the separability of sample points in the RGB and HSV color spaces to improve the understanding of the HSV transformation.

### 3.3. Urban Water Extraction Algorithm (UWEA)

The UWEA was designed as a three-step method based on an efficient water index and the advantage of HSV transformations (Figure 3). In the first step, the SWIR-blue space was employed to eliminate bright roofs. In the second step, the saturation-NIR space was used to eliminate vegetation via the HSV transformation of the SWIR1, NIR, and red bands. In the third step, a relatively dark impervious surface and shadow pixels were eliminated from the saturation-mNDWI space. Figure 4 depicts the water extraction process of the UWEA based on the sample dataset. The development of the UWEA was guided by three main considerations: (a) spectrum separability across different land cover types, (b) threshold or segmentation line robustness, and (c) multi-band combination. The thresholds in each step were determined iteratively. Single spectral bands, mNDWI, and saturation of the HSV color space were used to build the UWEA.

The majority of bright roof points (97.26%) were removed in the first phase of the UWEA because of the high value in both the blue and SWIR1 bands, with no loss of water points. In the second phase, NIR was employed to distinguish between water and vegetation points, and saturation aided in the separation of the overlapping points of these two classes. The thresholds in the first two phases were relatively flexible for retaining water points and removing non-water points. In the third step, the mNDWI and saturation band were used to separate the water and remaining non-water points, which had a horn-shaped distribution, that is, the water points spread in the top-right corner with a higher mNDWI and higher saturation, whereas the non-water points spread in the bottom-left

corner with a lower mNDWI and lower saturation; this distribution allowed for robust segmentation. The threshold in the third step was set to be relatively strict to obtain a clear final water map. However, the third step inevitably resulted in a water loss of 28.6% because it was very difficult to perfectly separate small rivers from the background with Landsat images. However, it would help to test which method can best extract small water bodies under complex urban landscapes. This segmentation was still more beneficial than using a single water index (e.g., mNDWI > 0.2). This was because it retained water points with a low mNDWI but high saturation (Figure 4e), which are often related to details or small water bodies in the water map. Notably, the UWEA substantially enhanced the separability of water and non-water objects by clustering and isolating different land cover types in the three UWEA segmentation spaces.

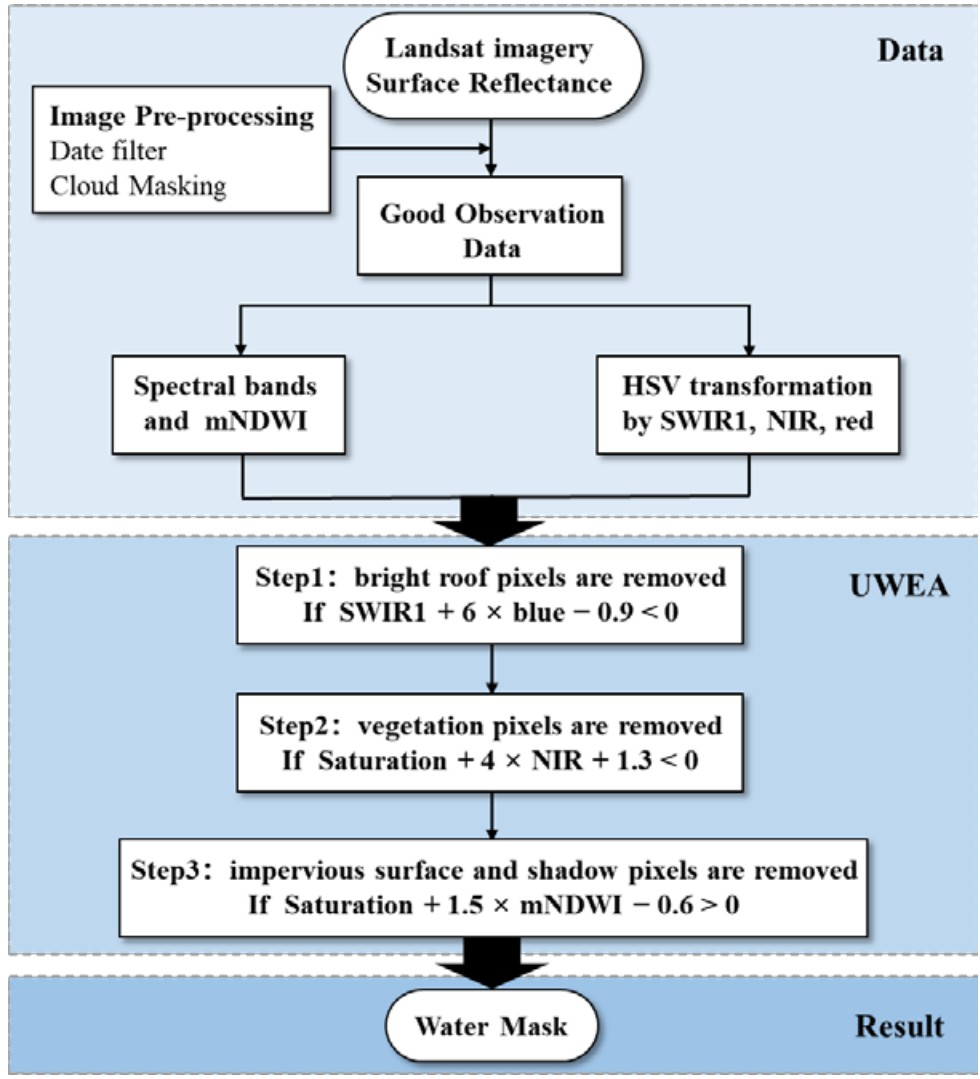

**Figure 3.** Flow chart of the UWEA water extraction process.

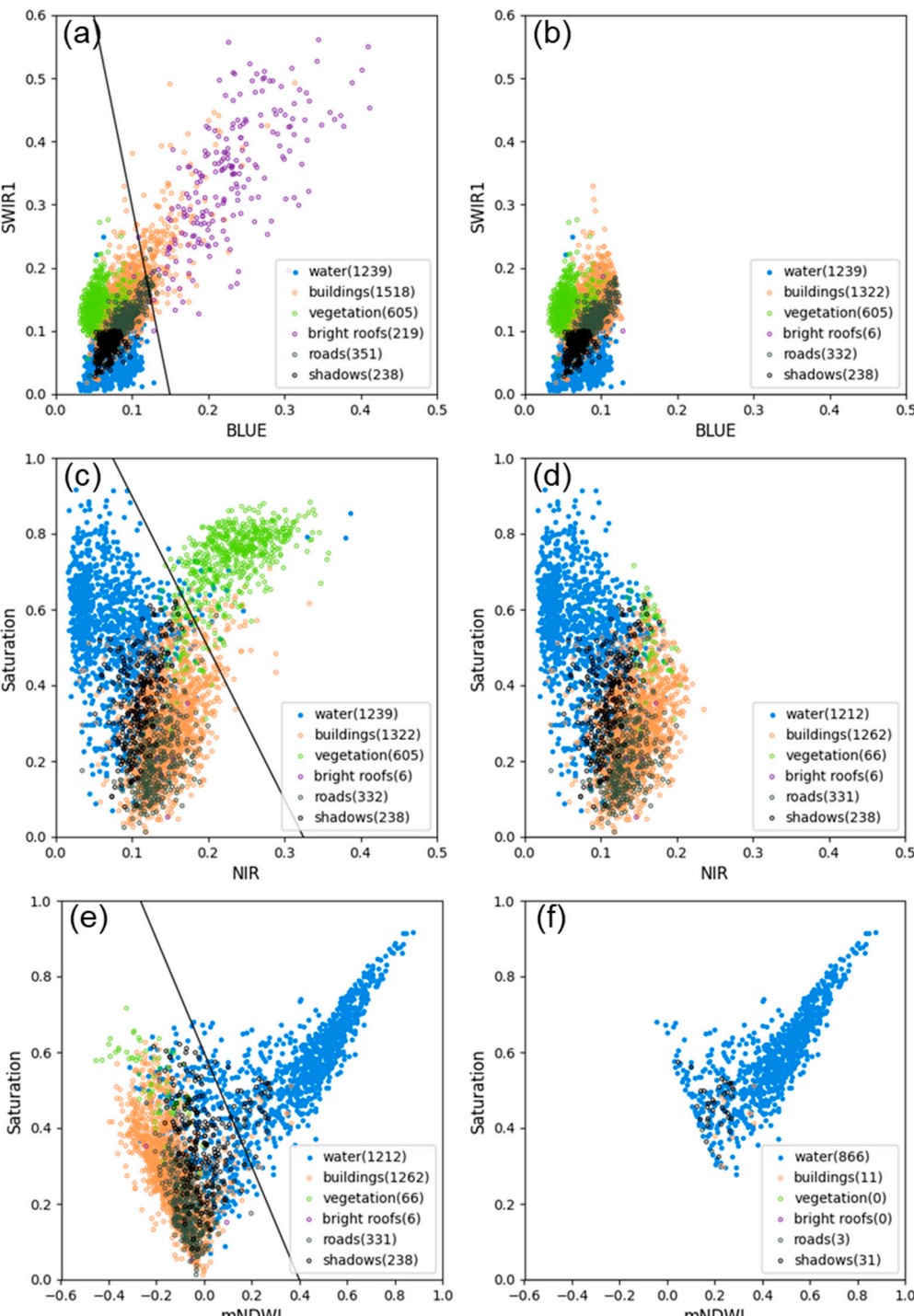

**Figure 4.** Segmentation of UWEA on sample points. The black line is the segmentation function of each step; (**a**,**c**,**e**) are the points of each land cover type before the segmentation, and (**b**,**d**,**f**) are the points that remained after the segmentation.

*3.4. Validation*

The accuracy of the UWEA was evaluated in the validation scenes and a comparison was made with the two commonly used urban water extraction methods, the mNDWI and the Jiang et al. [33] method (denoted as the HIS method). Using the sample dataset, the thresholds of the two contrast methods were optimized. The mNDWI threshold was set at 0.2. The HIS method involved two steps: (a) $210° < H < 270°$ and $S > min (S_{water})$ was used by the HIS transformations of the normalized NDBI, NDVI, and mNDWI to delineate water,

and (b) H < 90° or H > 150° and I > min ($I_{water}$) was used by the HIS transformation of the NIR, NDVI, and blue to eliminate shadows. $S_{water}$ denotes the saturation of water and $I_{water}$ represents the intensity (value) of water. The HIS method thresholds were optimized to min ($S_{water}$) = 0 and min ($I_{water}$) = 0.1. The threshold for New York was adjusted to 210° < H < 320° because of the Landsat-7 scan-line correction effect.

In each validation scene, two subregions (R1 and R2) were selected to compare the results of the different methods (Figure 5). True water maps were manually digitized from high-resolution images (Table 1). Three rates, namely the commission error, omission error, and total error, were used to assess the accuracy of the different approaches in each subregion.

$$\text{commission error} = \text{commission area}/\text{true water area} \tag{7}$$

$$\text{omission error} = \text{omission area}/\text{true water area} \tag{8}$$

$$\text{total error} = \text{commission error} + \text{omission error} \tag{9}$$

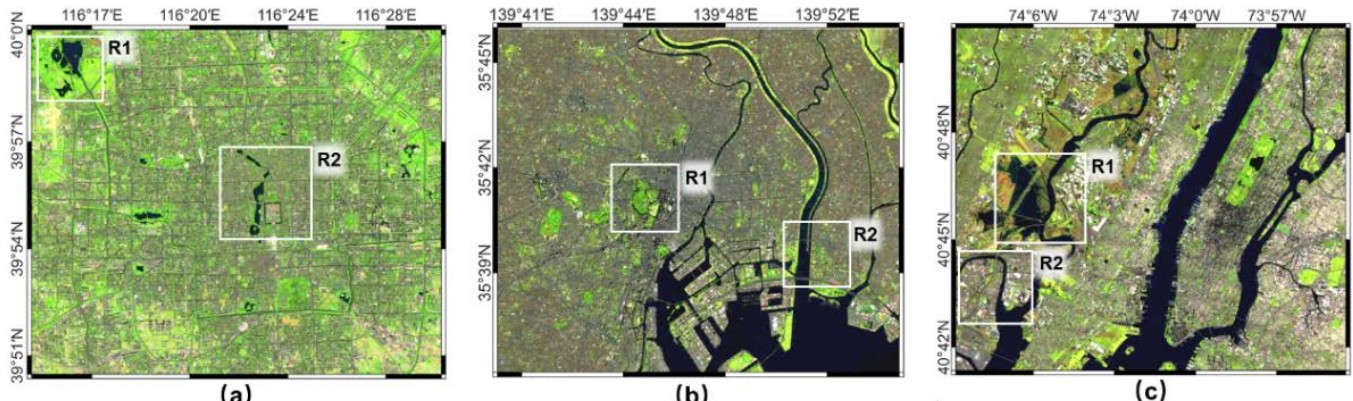

**Figure 5.** Selected subregions (R1 and R2) for accuracy assessment in each validation city, (**a**) Beijing, (**b**) Tokyo, and (**c**) New York. The false color image was composed using the SWIR1, NIR, and red bands.

In Equations (7) and (8), the commission, omission, and true water areas indicate the number of inaccurate extraction pixels, undetected water pixels, and reference water map pixels, respectively.

## 4. Results

### 4.1. Separability of HSV

The M-statistic test (*M*) was used to examine the separability of water and non-water samples in the HSV and RGB color spaces and the results are depicted in Figure 6. After conducting the HSV transformation from the RGB color space, the *M* between the water and the roads or shadows increased. Roads and shadows exhibited poor reflectance in all spectral bands, similar to water. However, for vegetation, buildings, and bright roofs, *M* decreased, particularly in water vs. vegetation. In some bands, these three land cover types had visibly greater reflectance than water. The HSV transformation decreased the separability of water from highly reflective objects while increasing the separability of water from low-reflective objects. Therefore, the UWEA employed spectral bands to eliminate highly reflective objects before performing the final segmentation using the HSV transformation.

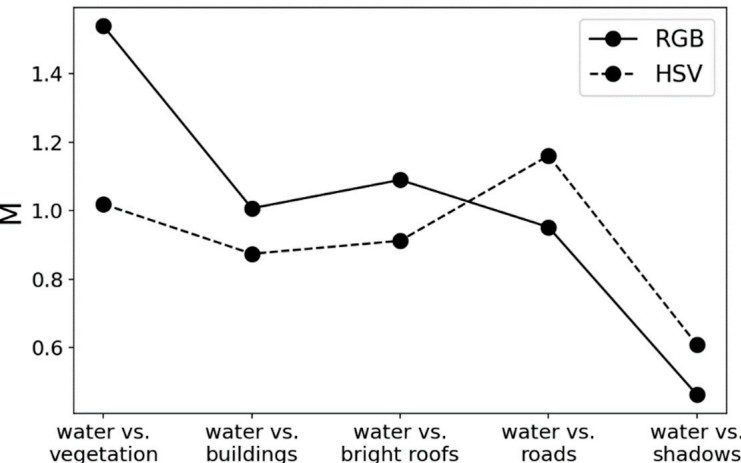

**Figure 6.** The M-statistic test of water and other land cover types in RGB and HSV color spaces, respectively.

*4.2. Water Mapping Results*

Figures 7–9 show the final water maps created using the three methods for the three validation cities. Overall, the visual inspection indicated that the UWEA approach was more accurate in urban water mapping than the mNDWI and HIS methods. For small water body detection, the UWEA demonstrated high sensitivity and effectively suppressed reflective impervious surfaces. The UWEA exhibited the best recall of small water bodies among the three approaches in the subregions of Beijing, as indicated by the light blue in Figure 7. In R1 of Beijing, inaccurate detections (red) of the UWEA were generated in part by errors in the manual digitization process since the border between water and vegetation was not clear in some places. In Tokyo (Figure 8), the UWEA and mNDWI suppressed shadows similarly, whereas the HIS method had more commission errors. In R1 of Tokyo, the UWEA still detected small water bodies better. In the subregions of New York (Figure 9), the UWEA improved the misidentification of reflective surfaces with the fewest missed detections of water bodies. Therefore, the proposed UWEA produced the best water extraction results compared to the other two methods.

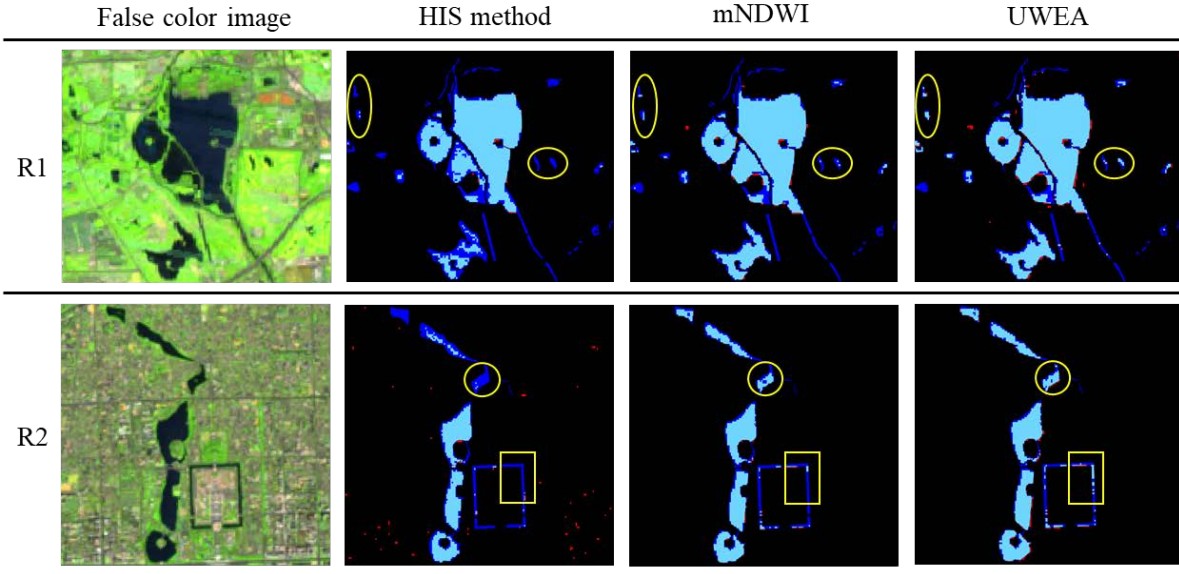

**Figure 7.** The water maps extracted via the HIS method, mNDWI, and UWEA for Beijing. The figures in light blue, dark blue, and red indicate accurately detected water bodies, undetected water bodies (omission area), and the inaccurate extraction of water bodies (commission area), respectively. The yellow circles and rectangles highlight the differences in extracted small water bodies using different methods.

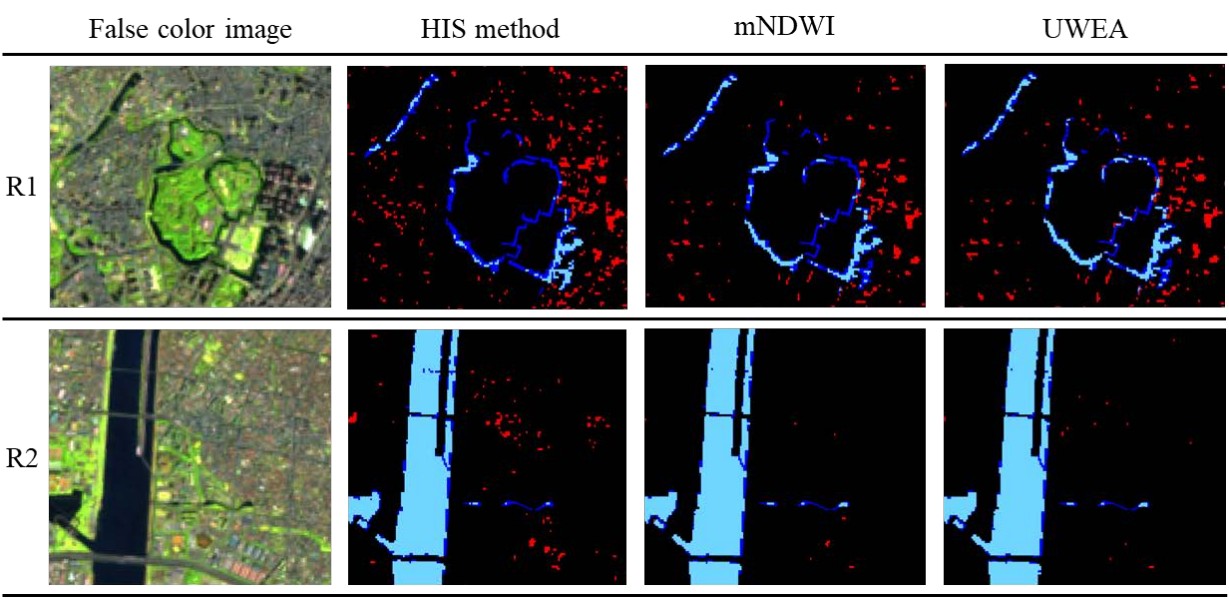

**Figure 8.** The same as Figure 7 but for Tokyo.

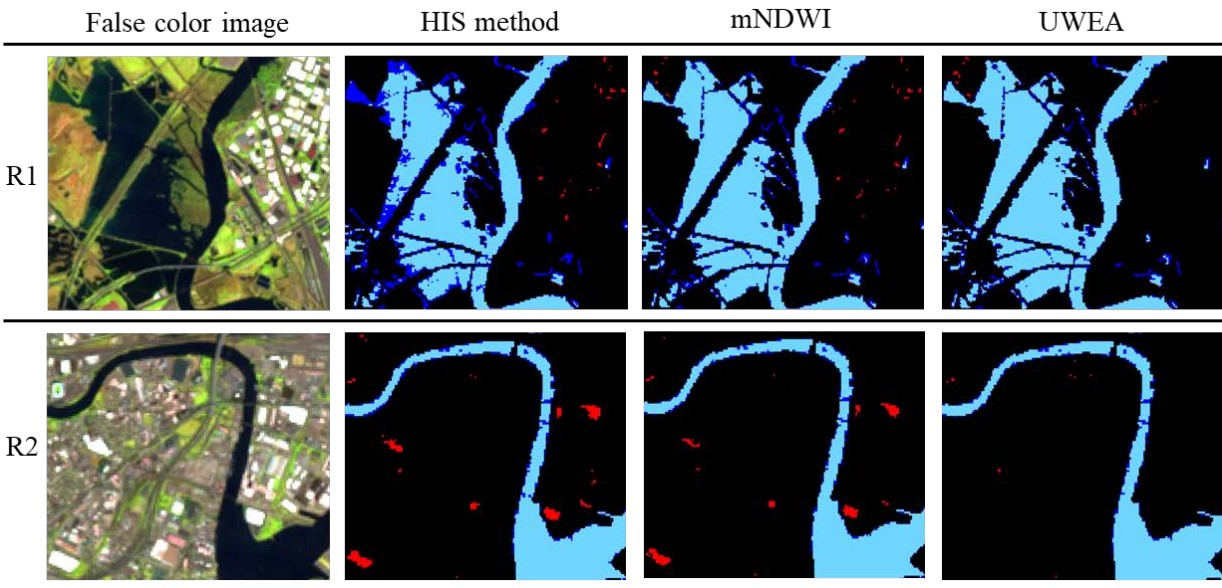

**Figure 9.** The same as Figure 7 but for New York.

### 4.3. Accuracy Assessment

Table 2 displays the classification accuracies of the three approaches in the subregions of the validation cities. On average, the UWEA method had the fewest total errors for all subregions, with over 3% fewer errors than the mNDWI and over 17% fewer errors than the HIS method. Furthermore, the omission errors of the UWEA were fewer than those of the other two methods in most cases, with the exception that they were approximately equal to the omission errors of the mNDWI in R2 of New York. In statistics, the omission errors of the UWEA were approximately 3% fewer than those of the mNDWI and 12% fewer than those of the HIS method and the differences were more pronounced in regions with small water bodies. The commission errors were not significant for any of the three methods. However, the UWEA commission errors in Beijing and Tokyo were greater than or equal to those of the mNDWI. The inaccurate detection in Beijing was not easy to distinguish since water and vegetation existed at the same time, but no evident mistakes in classifying vegetation as water were found for the three methods. In Tokyo, higher commission errors

mostly occurred in densely built areas with extremely dark shadows. In New York, the UWEA commission errors in the two subregions were both fewer than those of the mNDWI and HIS methods.

**Table 2.** The classification accuracy of the three methods for the validation scenes.

| Region | Method | Commission Error | Omission Error | Total Error |
|--------|--------|------------------|----------------|-------------|
| **Beijing** | | | | |
| R1 | UWEA | 3.05% | 28.85% | 31.89% |
| | mNDWI | 1.27% | 35.34% | 36.61% |
| | HIS method | 3.54% | 50.91% | 54.45% |
| R2 | UWEA | 2.69% | 23.12% | 25.81% |
| | mNDWI | 1.47% | 26.16% | 27.63% |
| | HIS method | 0.36% | 37.90% | 38.26% |
| **Tokyo** | | | | |
| R1 | UWEA | 38.52% | 52.46% | 90.98% |
| | mNDWI | 34.21% | 60.46% | 94.66% |
| | HIS method | 63.22% | 71.51% | 134.73% |
| R2 | UWEA | 0.42% | 9.68% | 10.10% |
| | mNDWI | 0.42% | 9.71% | 10.13% |
| | HIS method | 4.83% | 10.87% | 15.70% |
| **New York** | | | | |
| R1 | UWEA | 0.27% | 12.92% | 13.19% |
| | mNDWI | 0.77% | 15.16% | 15.93% |
| | HIS method | 0.77% | 25.95% | 26.72% |
| R2 | UWEA | 0.34% | 8.45% | 8.79% |
| | mNDWI | 5.84% | 8.37% | 14.21% |
| | HIS method | 7.64% | 9.89% | 17.53% |

The overall accuracies for the whole cities of Beijing, Tokyo, and New York were calculated and are shown in Table 3. The total study area of all three cities is about 500 km$^2$. The water area is 12.7 km$^2$ in Beijing, 71.82 km$^2$ in Tokyo, and 84.06 km$^2$ in New York. These are large in scale in comparison with the six selected validation regions. The calculated accuracies of the three methods were similar to those of the previous analysis in Table 2. For the total errors, the UWEA was superior to the mNDWI, which was superior to the HIS method. The omission errors accounted for a major portion of the total errors, and the omission errors of the UWEA were the fewest among the three methods for the three cities. For the commission errors, the UWEA was lower than the others for New York but higher than the mNDWI for Beijing and Tokyo. However, the differences between the accuracies of the three methods for the whole cities were smaller than those for the six validation scenes because wide rivers were easy to distinguish. Therefore, the UWEA had the fewest total errors for both the validation scenes and the whole cities.

In summary, the three methods performed equally in terms of commission errors, whereas the UWEA performed the best among the three methods in terms of omission and total errors. The advantages of the UWEA include the ability to detect small water bodies accurately and suppress reflective surfaces.

**Table 3.** The classification accuracy for the whole city.

|  |  | Beijing | Tokyo | New York |
|---|---|---|---|---|
| **Water Area (km$^2$)** |  | **12.70** | **71.82** | **84.06** |
| UWEA | Commission error | 4.42% | 2.27% | 0.73% |
|  | Omission error | 51.68% | 12.53% | 12.14% |
|  | Total error | 56.10% | 14.79% | 12.88% |
| mNDWI | Commission error | 1.94% | 2.01% | 1.15% |
|  | Omission error | 57.06% | 12.82% | 12.94% |
|  | Total error | 58.99% | 14.83% | 14.10% |
| HIS method | Commission error | 4.34% | 7.45% | 1.95% |
|  | Omission error | 66.90% | 12.31% | 14.73% |
|  | Total error | 71.23% | 19.76% | 16.69% |

*4.4. Robustness of Thresholds*

Setting thresholds for water indices is difficult because the signals of water and non-water pixels are always entangled. A better recall and fewer incorrect extractions cannot co-occur, and a moderately flexible threshold for detecting small water bodies will certainly result in more misidentifications of non-water objects. To further investigate the applicability of the UWEA in varied urban environments, the robustness of the default thresholds was examined and compared with the mNDWI and HIS methods. The default thresholds may be used as reference thresholds when implementing the UWEA in a new region; hence, robust thresholds can vastly minimize the trial-and-error period. Figure 10 depicts the variance in the total errors when adjusting the thresholds of the three methods for Beijing, Tokyo, and New York City. The UWEA and mNDWI had elementary change units of 0.05 and the UWEA changed in the third step. Because of the high number of omission errors, the default thresholds of the first step were magnified to $210° < H < 360°$ for the HIS method, and changes were added to the initial value ($210°$) with an elementary change unit of $10°$.

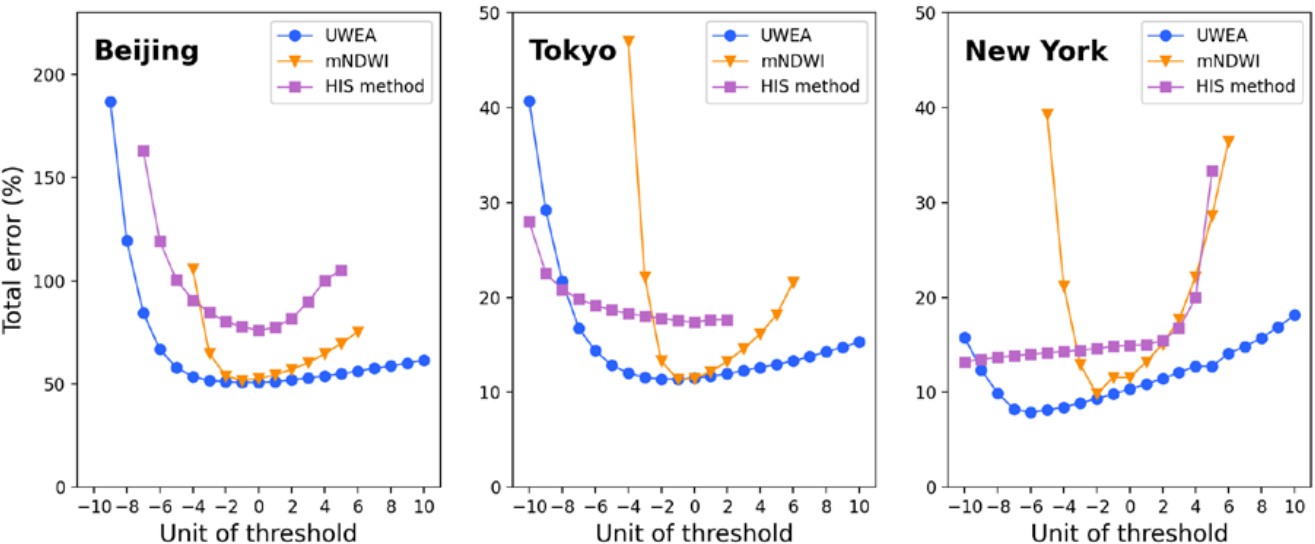

**Figure 10.** Changes in the total errors of three methods with thresholds for Beijing, Tokyo, and New York. The unit of threshold refers to 0.05 for the UWEA and mNDWI and $10°$ for the HIS method.

## 5. Discussion

### 5.1. Different Bands for HSV Transformation

Different band combinations can affect the performance of the HSV transformation. Band combinations in the literature include red–green–blue, NDBI–NDVI–mNDWI, and SWIR1–NIR–red. In general, bands that are sensitive to the differences between water and non-water objects benefit more from an HSV transformation. Figure 11 depicts the differences between the various color composite images for a scene in Beijing. The separability of water in the red, green, and blue bands is poor, as previously mentioned, and the abnormal rise in the NIR might cause confusion between water and vegetation when using the NDVI. Therefore, it is difficult to distinguish water from the background, as shown in Figure 11a,b. In contrast, water bodies were easily distinguished in the SWIR1–NIR–red composite image. In particular, a narrow river is distinguishable in Figure 11c. Using the SWIR1, NIR, and red bands has two advantages. The first is that water has consistently strong absorption in all three bands; hence, it has the lowest value and a dark color, and small water bodies are also visible in the composite map. The second advantage is that a low SWIR1 value results in the strong saturation and hue of water, which distinguishes it from impervious surfaces. Therefore, the HSV transformation based on the SWIR1, NIR, and red bands was more sensitive to small water bodies than the NDBI, NDVI, and mNDWI.

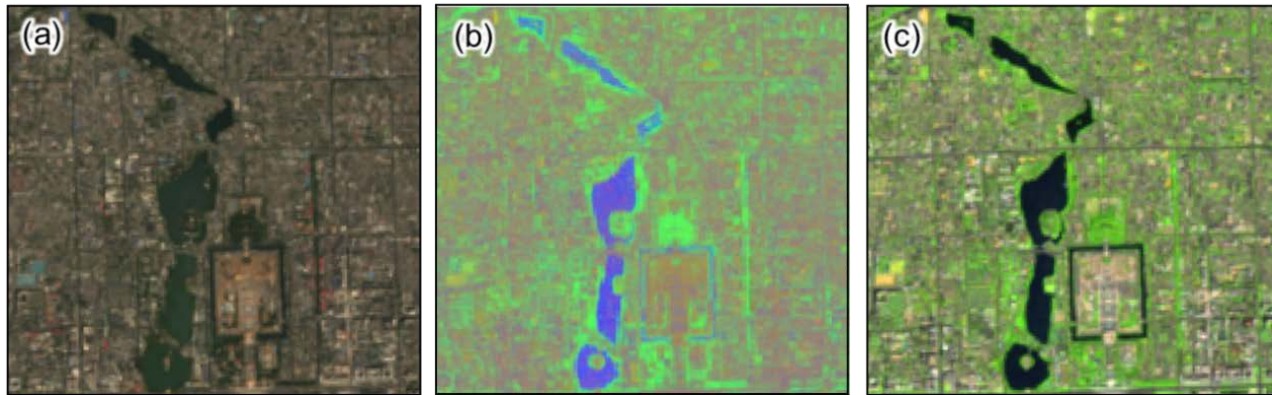

**Figure 11.** A scene in Beijing under different color composite images: (**a**) red–green–blue, (**b**) NDBI–NDVI–mNDWI, and (**c**) SWIR1–NIR–red.

### 5.2. Separability of the UWEA

The ability of the water extraction method to isolate target objects from background noise determines its applicability. Water index frequency histograms, which have bimodal distributions, can generally characterize separability, with great inter-group variances representing good separability. Thus, the Otsu method was applied to the three methods to perform a comparative analysis of separability. The Otsu method is an automatic threshold selection approach that maximizes the variances between clusters in a gray-level histogram [41]. Tokyo was the test site and all pixels in the region were used. In the first phase of the HIS method, the Otsu method was applied because H is the most important factor in water delineation. Because H is modulo 360° and there are two thresholds to consider, Otsu was conducted twice in the HIS method. For the UWEA, the Otsu method was applied in the third step. The revised thresholds were $153° < H < 284°$ for the HIS method, mNDWI > 0.23, and saturation +1.5 × mNDWI-0.6 > 0.24 for the UWEA. The Otsu approach and the water maps based on the recalculated thresholds are depicted in Figure 12. The three methods reserved the majority of the main rivers, although they differed in their misidentification of impervious surfaces. The HIS method resulted in more inaccurate urban area extractions than the mNDWI and UWEA methods. Although the UWEA produced similar results to the mNDWI, its threshold was more robust since the peaks of the water pixels (right) and non-water pixels (left) have a larger distance between them in the histogram (Figure 12e). Based on the new threshold, the M-statistic test results

for the three methods were calculated as follows: 4.19 for the UWEA, 3.24 for the mNDWI, and 3.95 for the HIS. The M-statistic test also revealed a better separability of the UWEA than those of the other two methods.

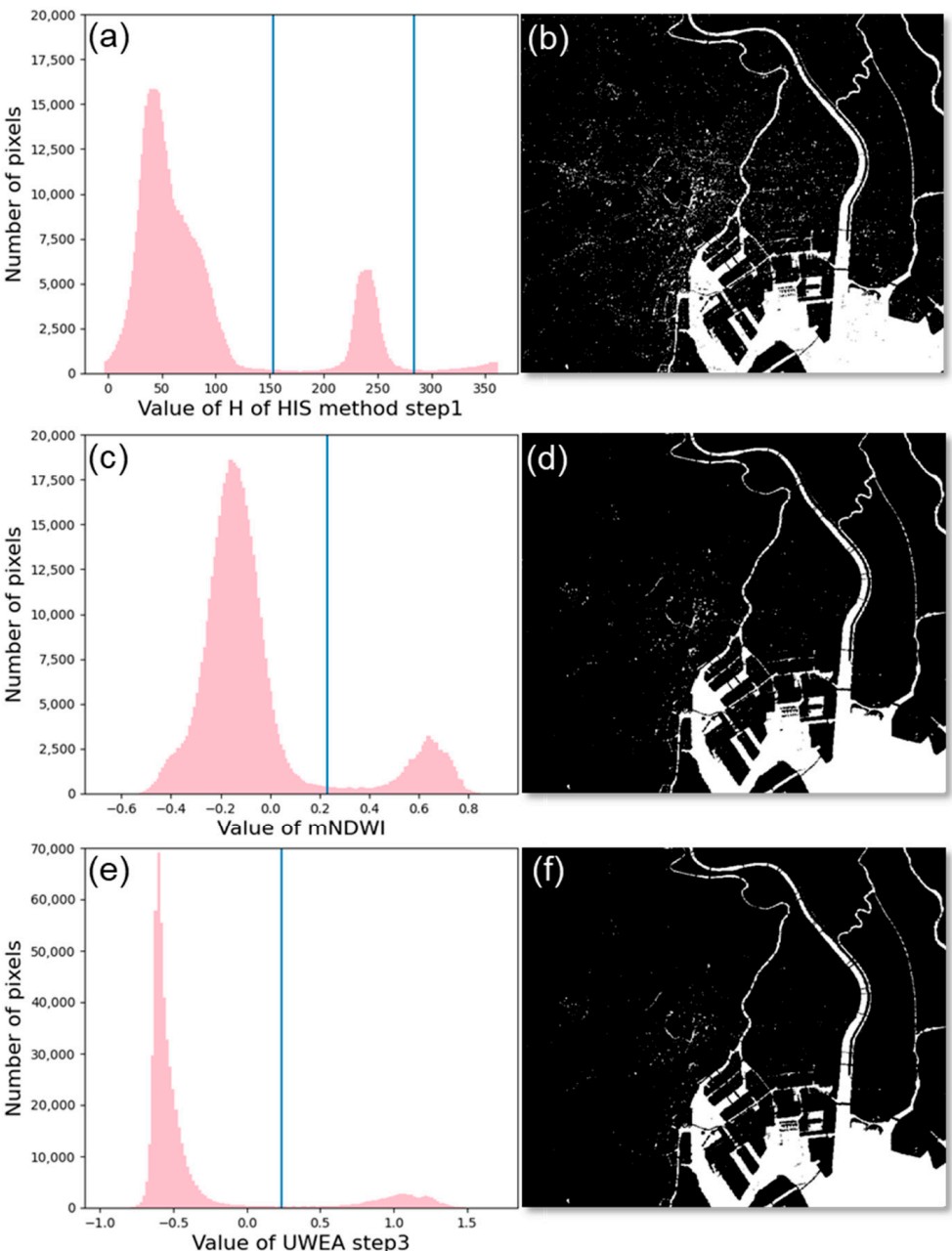

**Figure 12.** Subplots (**a**,**c**,**e**) are the frequency histograms of H of the first step of the HIS method, mNDWI, and the third step of the UWEA, respectively. The blue line is the threshold produced by the OTSU. Subplots (**b**,**d**,**f**) are the extracted results of the HIS method, mNDWI, and UWEA, respectively.

### 5.3. Advantages and Uncertainties

In the validation cities, lots of small rivers were digitized for validation, which was a challenge for these methods. Few studies have tested the water extraction methods in this level of detail. We found that only one river had a width of more than 3 times the image resolution that could be extracted and most methods were insensitive to rivers with small widths. However, there are many small rivers in urban areas. Small water body detection is currently one of the most challenging and urgent tasks. To identify water in urban areas, the UWEA employed multi-band information and combined the saturation of the HSV

color space with the mNDWI. Compared with the mNDWI and HIS methods, the UWEA approach had the fewest total errors for all the validation regions. This is particularly important for detecting small water bodies. In the HSV color space, which consists of the SWIR1, NIR, and red bands, water was more distinguishable with dark colors and high saturation. The UWEA used the mNDWI-saturation space to distinguish between water and non-water pixels. This method effectively eliminated impervious surfaces while retaining water with a low mNDWI but high saturation. This is why the UWEA was sensitive to small water bodies.

However, the UWEA exhibited no greater ability than the mNDWI in differentiating between water and extremely dark shadows. This is because the spectral characteristics of small water bodies and shadows are easily confused. A comparison of the accuracy assessment results indicated that omission errors rather than commission errors dominated urban water extraction. Although extremely dark shadows were present, the UWEA achieved fewer total errors by improving the omission errors. The UWEA is intended for urban water extraction and its accuracy in scenarios outside urban areas may be questionable. In terms of cloud-contaminated images, the first step of the UWEA can eliminate heavy clouds to some extent, but for thin clouds, the UWEA is as effective as the other methods. Because the mismatch between the annually composed images and the paired high-resolution images was avoided by visual examination, the residual uncertainties had a minor influence on the outcomes.

## 6. Conclusions

Surface water is a major land cover type in urban areas, with important implications for both the natural environment and human society. Accurate urban water mapping is critical for urban water resource management and water-related research of recent scenarios, given the increasing rate of urbanization and its ecological effects. However, the widespread presence of small water bodies and complex impervious surface compositions in urban areas pose a challenge to traditional water indices. In this study, an innovative algorithm (UWEA) was proposed for urban water detection using Landsat imagery, which combined spectral bands, the mNWDI, and the saturation of the HSV transformation. The HSV transformation was conducted using the SWIR1, NIR, and red bands, which proved to be sensitive to low-reflective objects in urban regions. As demonstrated in three cities with diverse urban characteristics, the UWEA had superior performance in small water body detection and reflective surface suppression and had the fewest total errors, which were approximately 3% less than the mNDWI and 17% less than the HIS method, on average. The default thresholds of the UWEA were robust in a variety of conditions. To the best of our knowledge, this is the first study that focuses on urban water extraction with an HSV transformation from Landsat data. The UWEA improves the accuracy of urban water mapping and allows city managers to access the spatiotemporal dynamics of urban water rapidly and accurately. This algorithm not only provides reliable results for urban water classification but also makes a significant contribution to other water-related studies.

**Author Contributions:** Conceptualization, L.C. (Liwei Chang), L.C. (Lei Cheng) and C.H.; methodology, L.C. (Liwei Chang); software, L.C. (Liwei Chang), C.F. and S.L.; validation, L.C. (Liwei Chang); data curation, L.C. (Liwei Chang), C.F. and S.L.; writing—original draft preparation, L.C. (Liwei Chang); writing—review and editing, L.C. (Lei Cheng), C.H. and S.Q.; supervision, L.C. (Lei Cheng) and S.Q.; funding acquisition, L.C. (Lei Cheng). All authors have read and agreed to the published version of the manuscript.

**Funding:** This research was funded by the National Natural Science Foundation of China (51961145104, 41890822).

**Data Availability Statement:** Publicly available datasets were used in this study. These data can be found at https://www.usgs.gov/landsat-missions/landsat-surface-reflectance (accessed on 21 February 2022).

**Acknowledgments:** This research was supported by the National Natural Science Foundation of China (51961145104, 41890822). We also appreciate the anonymous reviewers.

**Conflicts of Interest:** The authors declare no conflict of interest.

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
