# Peer review of "Extracting Urban Water Bodies from Landsat Imagery Based on mNDWI and HSV Transformation"

_remotesensing, doi:10.3390/rs14225785_

Round 1
Reviewer 1 Report
1. In this study, UWEA is used to separate water bodies in a similar decision tree classification method, which is indeed superior to HIS method and mNDWI method in terms of results.
2. It is necessary to discuss the advantages and disadvantages between the image classification method and this research method in the Introduction chapter, and propose why the method of this research is used.
3. If there is a defect in the non-water area in the initial filtration, and there is a misjudgment in the initial filtration, is there a remedial mechanism?
4. Figure 4 shows the segmentation results of sample points using mNDWI, but the separation effect does not seem to be good. Do you consider image conversion methods such as PCA and MNF to improve the separation of spectral values?
5. Please analyze whether the water body and shadow have sufficient spectral separation in blue, green or other bands?
6. Although the image sampling of the water body avoids the influence of mixed pixels, how to solve the mixed pixel problem of buildings, bright roofs and roads?
7. Please use different images such as SPOT7 or FM5 satellite images with higher spatial resolution to see if the same results can be obtained.
8. In the test groups of R1 and R2 in Figure 7, the results of MNDWI and UWEA images are almost the same. Please indicate the difference in total error between 31.89% and 36.61% in Table 2 on the result map.
9. The research must provide the error amount of each category. Except the total error of New York R2 UWEA is less than 10%, almost all other groups are greater than 10%. To be honest, the accuracy rate needs to be strengthened.
Author Response
We really appreciate your suggestions. We have modified all the comments you mentioned. Please see the attachment.

Reviewer 2 Report
Overall - great job!
Fig. 10 - poor quality and over the caption of Fig. strange grey line.
As a specialist i know what kind of software you have been using. But for the reader - please provide one sentense about it.
Author Response

(The authors gave the same response as above.)

Reviewer 3 Report
The manuscript ‘Extracting urban water bodies from Landsat imagery based on mNDWI and HSV transformation’ submitted to Remote sensing requires minor improvements and resubmission before it can be considered for publication. I found the following drawbacks of the study:
1. Author stated that UWEA method aims to enhance the accuracy of urban water extraction as compared to prevalent methods, whereas there is only a marginal improvement of 3% in error as compared to mNDWI.
2. Results for only validation are presented. The total number of urban water bodies mapped using UWEA in the selected six cities should be provided. A comparison with the existing total number of urban bodies in the selected cities will give a better idea of the method's applicability.
3. Total error for Tokyo (90.98%) in the validation for R1 and for R2 is only 10.10%. R1 and R2 are selected from the satellite imagery for Tokyo. Then why is there so much error?
Specific Comments are provided in the attached pdf with highlighted text and comments.

Author Response

(The authors gave the same response as above.)
